# A Comparison of In-Person and Telehealth Personalized Exercise Programs for Cancer Survivors: A Secondary Data Analysis

**DOI:** 10.3390/cancers17152432

**Published:** 2025-07-23

**Authors:** Nada Lukkahatai, Gyumin Han, Chitchanok Benjasirisan, Jongmin Park, Hejingzi Monica Jia, Mingfang Li, Junxin Li, Jennifer Y. Sheng, Michael Carducci, Leorey N. Saligan

**Affiliations:** 1School of Nursing, Johns Hopkins University, Baltimore, MD 21205, USA; cbenjas1@jh.edu (C.B.); junxin.li@jhu.edu (J.L.); 2College of Nursing, Research Institute of Nursing Science, Pusan National University, Yangsan 50612, Republic of Korea; hangyum@pusan.ac.kr (G.H.); jmpark@pusan.ac.kr (J.P.); carducci@jhmi.edu (M.C.); saliganl@mail.nih.gov (L.N.S.); 3Department of Epidemiology, Johns Hopkins Bloomberg School of Public Health, Baltimore, MD 21205, USA; jia00176@umn.edu; 4School of Nursing, Sun Yat-sen University, Guangzhou 510275, China; limfsysu123@gmail.com; 5School of Medicine, Johns Hopkins University, Baltimore, MD 21205, USA; jsheng7@jhmi.edu; 6National Institute of Nursing Research, National Institutes of Health, Bethesda, MD 20892, USA

**Keywords:** personalized exercise, telehealth, symptom science, cancer outcomes

## Abstract

This study compared the effectiveness of a personalized 12-week exercise program delivered in-person versus via telehealth among cancer survivors experiencing symptoms like fatigue, pain, and cognitive impairment. Drawing on secondary data from two randomized controlled pilot studies, the research aimed to assess changes in symptoms, resilience, and quality of life, and to evaluate whether telehealth could be a viable alternative to in-person delivery. Among 75 participants, those who received either form of the exercise intervention reported significant improvements in physical and mental fatigability, while the control group showed worsening pain. No significant differences emerged between delivery methods for most outcomes, except for slightly better improvement in visuo-perceptual cognitive difficulty in the telehealth group. The findings suggest that telehealth-based personalized exercise programs can be as effective as traditional in-person approaches, offering a scalable and accessible solution for cancer survivors, particularly those with mobility challenges or limited access to care. This research supports the integration of telehealth into survivorship care, potentially reducing symptom burden and improving recovery in a cost-effective and patient-centered manner. Further large-scale studies are needed to confirm these results and address barriers such as digital literacy and access.

## 1. Introduction

Advancements in cancer treatment have significantly improved survival rates of cancer patients, bringing post-treatment symptoms to the forefront of concerns affecting cancer survivors. Cancer-related symptoms such as fatigue, pain, cognitive impairment, and sleep disturbance profoundly impact health-related quality of life (HRQOL), daily activities, and overall well-being [1]. Fatigue, in particular, affects approximately 50–60% of cancer survivors who have undergone anticancer treatment, necessitating strategies to improve and maintain their daily lives [2,3]. Despite these advancements, there remains a gap in our understanding of these symptoms’ full scope and impact on the lives of cancer survivors.

Regular exercise has been shown to significantly improve cancer-related symptoms, including fatigue and muscle weakness, promoting overall well-being and physical strength, and alleviating various side effects of cancer treatments [4,5]. These effects are supported by both physiological mechanisms, such as reductions in systemic inflammation, neuroendocrine modulation, and increased neurotrophic factors, [6,7] and psychological pathways, including improved mood, self-efficacy, and emotional regulation [8,9]. Importantly, exercise also plays a key role in supporting health-related quality of life (HRQOL), a multidimensional concept that encompasses physical, psychological, and social well-being in the context of health status [10]. Additionally, resilience, or the capacity to adapt positively to health-related adversity, is increasingly recognized as a protective factor that can enhance coping, engagement in self-care behaviors, and emotional adjustment following cancer treatment [11,12]. Despite the benefits of exercise, most cancer patients fail to meet physical activity (PA) recommendations due to: (1) physical limitations and burden, and (2) limitations in access to an exercise facility and scheduling conflicts [13,14,15]. To overcome these barriers to exercise, many studies have reported the benefit of personalized home-based exercise interventions for cancer recovery and self-management of cancer-related symptoms [1,16,17,18,19,20,21].

Guided by the National Institutes of Health (NIH) Symptom Science Model, which emphasizes multidimensional symptom assessment and targeted intervention, we developed the Personalized Home-Based Exercise Program (iHBE) (ClinicalTrials.gov: NCT03874754) to pilot test the feasibility of an individualized, home-based exercise approach. The iHBE study utilized in-person home visits and telephone follow-up support to address the diverse and interrelated symptoms experienced by cancer survivors. Designed to improve adherence, alleviate common symptoms, and enhance quality of life, the iHBE intervention was grounded in the NIH Symptom Science Model, which systematically characterizes, monitors, and addresses cancer-related symptom burden through tailored strategies delivered directly to participants in their homes. In line with the model’s emphasis on patient-centered outcomes, the intervention also targeted improvements in HRQOL and resilience, both of which are increasingly recognized as critical indicators of functional recovery and well-being among cancer survivors.

During the COVID-19 pandemic, about 20% of cancer survivors decreased their willingness to engage in in-person exercise activities due to health risk concerns and ongoing health crises [22]. To address these concerns, telehealth approaches, including the use of smartphone-based applications, were used in multiple healthcare settings and services [23]. In response to these changing circumstances and building on our experience with the original iHBE study, we adapted the intervention to develop the Technology-Enhanced Home Exercise Program (TEHE). The TEHE study, guided by the NIH Symptom Science Model, utilized a smartphone application to deliver weekly exercise recommendations and monitor participants’ symptoms and activity levels remotely (ClinicalTrials.gov: NCT03576274). This innovation allowed for continued personalized support while prioritizing the safety of cancer survivors.

Multiple studies have reported promising effects of exercise programs delivered through telehealth technology on physical function and general symptoms [24,25,26,27]. However, telehealth exercise programs come with limitations that limit their potential effectiveness, including the potential difficulty in maintaining patient motivation, the inability to provide the same level of personalized encouragement, and the challenge of accurately assessing an individual’s physical condition [28,29]. Although both in-person and telehealth-based exercise interventions have shown benefits in various studies, few investigations have explored how these delivery modalities might yield different outcomes for cancer survivors, particularly when using similar intervention content and measurement across settings. A recent mixed-methods study highlighted the distinct advantages and challenges of both formats from participants’ perspectives and emphasized the need for future trials that explicitly compare delivery modalities with respect to outcomes like adherence, satisfaction, and symptom management [30].

While the small sample sizes and pilot nature of the iHBE (in-person) and TEHE (telehealth) studies preclude definitive conclusions, combining data from these two methodologically similar pilot trials presents a valuable opportunity for exploratory, delivery-modality comparison. Such an analysis can generate preliminary insights into the potential advantages and limitations of each approach, help identify trends or practical considerations for future research, and provide important groundwork for designing larger, adequately powered trials that directly evaluate different exercise delivery methods in cancer survivorship.

Based on prior evidence and our intervention design, we hypothesized that participants receiving the personalized exercise programs (iHBE or TEHE) would demonstrate improvements in cancer-related symptoms (including fatigue, pain, sleep, and cognitive function), physical function, resilience, and HRQOL from baseline to post-intervention. We also expected that the telehealth-delivered intervention (TEHE) would yield comparable outcomes to the in-person intervention (iHBE), reflecting the feasibility and potential of remotely delivered exercise programs for cancer survivors.

Therefore, this paper aims to: (1)evaluate the effectiveness of personalized exercise programs on cancer-related symptoms (e.g., pain, fatigue/fatigability, sleep, cognitive function, etc.), physical function, resilience, and HRQOL and,(2)compare the impact of the program when delivered through in-person sessions versus telehealth.

## 2. Materials and Methods

### 2.1. Study Design

This is a secondary, exploratory comparative analysis using data from two 12-week randomized controlled pilot studies designed to examine personalized exercise interventions in cancer survivors. The two studies were similar in duration, outcome measures, participant eligibility, and assessment timing, allowing for integration of data to compare intervention delivery modalities. The first study, the Personalized Home-Based Exercise Program (iHBE), involved in-person home visits for the first three weeks and nine telephone follow-ups (ClinicalTrials.gov: NCT03874754). The second study, the Technology-Enhanced Home Exercise Program (TEHE), used a smartphone application to deliver weekly exercise recommendations and interventions (ClinicalTrials.gov: NCT03576274).

### 2.2. Participants

For this secondary analysis, we purposefully selected participants from the exercise intervention groups and control groups of two pilot studies—iHBE and TEHE studies—to evaluate and compare outcomes associated with different modes of exercise intervention delivery. Both studies used a convenience sampling approach, recruiting eligible participants from oncology outpatient clinics affiliated with a comprehensive cancer center in Baltimore, Maryland, USA. Recruitment was conducted through a combination of posted flyers within the cancer center, social media postings, and targeted electronic health record (EHR) outreach via the MyChart EPIC recruitment service. Potential participants identified through MyChart were invited electronically, while individuals who saw the flyers and social media posts self-referred to the study. In total, two participants in the iHBE study and four participants in the TEHE study enrolled via self-referral. Recruitment for the parent studies occurred from 2018 to 2022, with participant follow-up completed by 2023. These are the periods during which data were collected for the present analysis. Eligible individuals met the following criteria: (1) aged 18 years or older; (2) survivors of solid tumor cancers; (3) experiencing an average fatigue rating of 3 or higher on a 10-point Likert scale over the past seven days; (4) able to communicate in English; (5) smartphone owners; (6) not diagnosed with a psychological disorder; and (7) having completed primary cancer treatment (surgery, chemotherapy, and/or radiation therapy). All study procedures, including consent, baseline and follow-up assessments, and intervention delivery, were conducted either in participants’ homes or remotely, depending on the respective study trial (in-person visits for iHBE; telehealth/app-based for TEHE). Both studies recruited participants from the same geographical area using similar recruitment procedures, ensuring comparability across the samples. Demographic and clinical characteristics of the selected sample are summarized in Table 1.

### 2.3. Assignment Method and Blinding

In the original iHBE and TEHE pilot studies, eligible participants were randomly assigned to either the intervention or control group using computer-generated randomization procedures. For the present secondary analysis, only participants originally assigned to the intervention groups in both studies were included.

Due to the nature of the interventions in the original iHBE and TEHE pilot studies, neither the participants nor the research staff delivering the interventions were blinded to group assignment. Most outcome data were collected via participant self-report and objective device recordings, and outcome assessors were not blinded. As this is a secondary analysis of two pilot studies, no additional blinding or masking procedures were implemented.

### 2.4. Interventions

All components of the iHBE and TEHE interventions—including in-person home visits, telephone calls, online sessions, and app-based communications—were delivered by research staff with training in exercise guidance for cancer survivors. The interventions were as follows.

#### 2.4.1. In-Person Home Exercise Program (iHBE)

This 12-week tailored home-based exercise program utilized mobile technologies, including a PA tracker and a mobile application to monitor participants’ PA, symptoms, and adherence to the exercise regimen, offered real-time feedback and sent daily reminder messages. The intervention consisted of one initial assessment home visit, two in-person home visits during the exercise phase, and nine follow-up telephone calls. During the three in-person home visits, the participants were introduced to the technology, including setting up exercise modalities (week 1); then, they were introduced to sample exercises, correct body alignment and composition, physical performance reviews, goal adjustment, and addressing challenges (weeks 2 and 3). Participants’ PA and symptoms were monitored daily through mobile technologies. The PA goals were adjusted weekly based on the weekly PA level and on participants’ self-reported goal achievements. Nine weekly (week 4 to 11) follow-up telephone calls were conducted to review the physical performance and challenges and adjust weekly goals. At the end of the 12-week intervention, the overall performance was evaluated, and long-term maintenance goals were discussed with each participant.

To increase compliance and adherence, participants received weekly feedback on their progress through in-person visits and telephone calls. These visits and follow-ups provided additional opportunities for personalized encouragement, troubleshooting barriers, and weekly adjustment of goals based on self-reported activity and achievement. The same interventionist conducted all in-person intervention visits and follow-up telephone calls for each participant throughout the 12-week program to ensure consistency and rapport.

#### 2.4.2. Technology-Enhanced Home Exercise (TEHE) Program-Telehealth

This 12-week Technology-Enhanced Home Exercise (TEHE) program utilized mobile technologies, including a PA tracker and a mobile application to monitor daily participants’ PA and symptom severity. The intervention included one initial online individual session and weekly exercise recommendations through a smartphone application. The initial online session covered goal setting, discussions of PA preferences and barriers, exercise safety, and body alignment. The smartphone application was used to send weekly PA recommendations, PA performance reports, a short survey for weekly goal achievement, and feedback for the recommendations. Following the recommendations of the Cancer Exercise Training [31] Institute and the American Cancer Society [32] that cancer survivors engage in regular PA tailored to their capabilities and limitations, the PA recommendations were adjusted weekly based on data from the PA performance reports and goal achievement surveys. At the end of the 12 weeks, the overall performance and long-term maintenance goals were reported and recommended to each participant.

To facilitate adherence in the TEHE group, ongoing support included automated daily reminders from the mobile application, weekly feedback based on tracked activity and symptom reports, a goal achievement survey at the end of each week, and continuous access to digital progress reports. The use of technology enabled participants to self-monitor, receive timely encouragement, and adjust their goals regularly, thereby maintaining motivation throughout the 12-week intervention. Each participant was paired with the same interventionist throughout the 12-week program to promote continuity and individualized support.

#### 2.4.3. Standard Care Control Group

Participants in the standard care control group for both the iHBE and TEHE studies continued with their usual PA. They were asked to wear a PA tracker and respond to a daily symptoms survey via a smartphone application while continuing their usual activity for 12 weeks.

### 2.5. Outcomes and Measures

#### 2.5.1. Primary Outcomes

*Pain* was evaluated using the Patient-Reported Outcome Measurement Information System (PROMIS) Short Form-Pain 3a v1.0 (PROMIS-Pain). This 3-item tool assesses an individual’s average pain level over the past seven days on a 5-point Likert scale, with responses ranging from 1 (no pain) to 5 (severe pain). The summary scores were converted to *T*-scores, where higher scores indicate greater pain levels. The Cronbach’s alpha coefficient for this measure ranged from 0.82 to 0.90 [33].

*Fatigue* was measured using the PROMIS Short Form v1.0-Fatigue 6a. This is a 6-item tool assessing self-reported fatigue in terms of frequency, duration, intensity, and its impact on physical, mental, and social activities, using a 5-point Likert scale from 1 (not at all) to 5 (very much). Summary scores were converted to *T*-scores, with higher scores indicating greater fatigue. The tool demonstrated strong internal consistency, with a reliability coefficient of greater than 0.80 [34].

*Fatigability* was measured by the Pittsburgh Fatigability Scale (PFS). The PFS is a 10-item tool that rates physical and mental fatigability on a 6-point Likert scale from 0 (no fatigue) to 5 (extreme fatigue). Higher total scores indicated higher levels of physical and mental fatigability. The PFS has strong internal consistency, with a Cronbach’s alpha coefficient of 0.88 [35].

*Sleep* was evaluated using the Insomnia Severity Index (ISI) and an average of 7 days of sleep duration. This 7-item ISI assessed sleep using a 5-point Likert scale. The total score was categorized into four groups:0–7 (no clinically significant insomnia)8–14 (subthreshold insomnia)15–21 (moderate clinical insomnia) and22–28 (severe clinical insomnia)

The ISI demonstrated acceptable reliability, with a Cronbach’s alpha coefficient of 0.73 [36].

*Objective sleep durations* were measured as a 7-day average calculated from daily recordings on the PA tracker during the 7 days before the intervention and the last 7 days of the intervention. Both studies used a commercially available PA tracking device, the Fitbit^TM^ Charge (Fitbit Inc., San Francisco, CA, USA) to track motion and PA data (e.g., steps and sleep duration.) The PA data were stored in a secure, web-based platform. This Fitbit device showed a good agreement with the physical measurement gold standard, the activPAL^TM^ (PAL Technologies Ltd., Glasgow, UK) in healthy participants during sleep and sedentary behavior, with an intraclass correlation coefficient = 0.94 (95% confidence interval: 0.92–0.96) [37].

#### 2.5.2. Secondary Outcomes

*Cognitive function* was measured using the Montreal Cognitive Assessment (MoCA) and the Multiple Ability Self-Report Questionnaire (MASQ). The MoCA is designed to detect comprehensive cognitive impairment across various cognitive domains, including recall, abstraction, naming, attention, language, memory, visuospatial/executive functions, and orientation to time and place. In cancer survivors, the MoCA demonstrated a Cronbach’s alpha of 0.79 [38]. The MASQ is a 38-item questionnaire covering five domains: language, visuo-perceptual ability, verbal memory, visual memory, and attention. Each item is rated on a 5-point Likert scale from 1 (never) to 5 (always). The total score for each domain ranges from 8 to 40, except for visuo-perceptual ability, which ranges from 6 to 30. Higher scores indicate greater perceived cognitive dysfunction in the respective domains. The MASQ subscales have shown high internal consistency, with Cronbach’s alpha coefficients ranging from 0.72 to 0.74 [39].

*Physical function* was measured using the Virtual Short Physical Performance Battery (vSPPB; developed by Wake Forest School of Medicine, Winston-Salem, NC, USA) and a 7-day average of steps measured by the PA tracker. The vSPPB is a digital adaptation of the widely used Short Physical Performance Battery (SPPB) designed to assess the individuals’ perception of gait speed, chair stand, and balance. Scores range from 0 (worst performance) to 12 (best performance), with acceptable test-retest reliability and results comparable to established physical performance measures [40,41].

*Objective physical activity level* was measured using a 7-day average of steps recorded daily on PA tracking device before the intervention and the last seven days of the intervention. Both studies used Fitbit^TM^ Charge (Fitbit Inc., San Francisco, CA, USA) to track motion and PA data. This Fitbit^TM^ device showed a good agreement with the physical activity measurement gold standard, the ActiGraph^TM^ GT3X (ActiGraph LLC, Pensacola, FL, USA), in healthy participants during treadmill walking and during the free-living environment, withintraclass correlation coefficients = 0.81 and 0.96, respectively (*p* < 0.01) [42,43].

*Resilience* was measured using the Connor–Davidson Resilience Scale, which includes 10 self-report items that rate participants’ opinions about their psychological resilience. Each item was scored on the 5-point Likert scale ranging from 0 (not true at all) to 4 (true nearly all the time). A higher total score indicates a better resilience condition. This instrument has good internal consistency, with a Cronbach’s alpha coefficient of 0.85 [44].

*Health-related quality of life* was measured using the Short-Form survey (SF-36). This 36-item self-report instrument measures the physical and mental psychological domains of HRQOL. Scores are calculated by summing responses across items and transforming these raw scores to a 0 to 100 scale, with higher scores indicating better function and health. The Cronbach’s alpha coefficient was 0.70, showing good internal consistency [45].

### 2.6. Data Collection Procedures

Data for this secondary analysis were obtained from two previously conducted pilot studies (iHBE and TEHE). For both studies, data on all primary and secondary outcomes were collected at baseline (prior to intervention) and immediately following the 12-week intervention period. Self-report questionnaires were administered electronically using the REDCap (Research Electronic Data Capture) system version 15.0.34 (Vanderbilt University, Nashville, TN, USA), which included built-in range checks and logic controls to enhance data accuracy and completeness. Objective physical activity and sleep data were continuously monitored using commercially available PA tracking devices and uploaded to a secure web-based platform. For selected measures (e.g., Montreal Cognitive Assessment: MoCA), assessments were conducted in-person or via telehealth by trained research staff following standardized protocols established in the original trials. Data quality was further supported by the use of validated instruments and routine oversight by study personnel. Full details of data collection methods and quality assurance procedures are described in the respective primary study protocols.

### 2.7. Statistical Analysis

The unit of analysis for all comparisons was the individual participant. Participants from two pilot studies were included in this analysis. They were categorized into three groups: an in-person home exercise group (intervention group from the iHBE study), a telehealth home exercise group (intervention group from the TEHE study), and a standard care control group (control groups from both studies). Because both studies used similar protocols, measurement timepoints, and outcome measures, the control groups were combined to increase statistical power and enhance analytic consistency. Descriptive statistics were used to summarize the participants’ baseline characteristics. To identify differences in baseline characteristics across the three groups (iHBE, TEHE, and controls), χ^2^ tests were conducted for categorical data, while *f*-tests were used for continuous data. A dependent *t*-test was used to examine the effectiveness of personalized exercise programs in alleviating symptoms such as pain, fatigue/fatigability, and sleep, as well as cognitive function, physical function, resilience, and HRQOL. This analysis compared the scores before the intervention (T1) to those after the 12-week intervention (T2) within each group.

An analysis of covariance (ANCOVA) was conducted to compare the program’s impact when delivered through in-person sessions versus telehealth. This analysis compared the mean changes in outcomes between the in-person and telehealth groups, adjusting for covariates including age and education, which showed notable baseline differences. Other sociodemographic variables such as marital status and employment status were not included to avoid overfitting due to the small sample size. Statistical significance was set at *p* < 0.05. All statistical analyses were performed using Statistical Package for the Social Sciences (Version 27) and Stata SE (Version 17, StataCorp, College Station, TX, USA). Analyses were conducted on all participants with available outcome data according to their original group assignment. Due to the secondary nature of this analysis and incomplete data for some participants, a strict intention-to-treat approach was not possible. No methods were used to impute missing data, and non-compliers were included if outcome data were present; otherwise, they were excluded from analyses of that outcome.

### 2.8. Sample Size

As this project is a secondary and exploratory analysis of data obtained from two pilot studies (iHBE and TEHE), no formal sample size calculation was conducted for the present analysis. The sample size was determined by the number of eligible participants enrolled in the intervention arms of the original studies. No interim analyses or formal stopping rules were implemented in the original pilot studies or this secondary analysis.

## 3. Results

### 3.1. Participant Flow and Baseline Characteristics

This secondary analysis included 75 participants with available data from the iHBE and TEHE pilot studies: 15 in the in-person exercise (iHBE) group, 38 in the telehealth exercise (TEHE) group, and 22 in the control group. Participants were included in these analyses if they had baseline and/or post-intervention data for at least one outcome of interest. The mean (±SD) age of participants was 64.59 (±13.43) years in the control group, 74.33 (±7.62) years in the iHBE group, and 58.61 (±12.10) years in the TEHE group (Table 2). Gender distribution was approximately equal across groups, with 50% of participants in the control group, 46.7% in the iHBE group, and 47.4% in the TEHE group being male. The racial composition showed significant differences, with a higher percentage of White participants in the TEHE group (84.2%) compared to the control (59.1%) and iHBE groups (33.3%). Educational levels varied, with the majority of the TEHE group (89.2%) having graduated from college, compared to 38.9% in the control group and 57.1% in the iHBE group. Employment status and marital status were similar across the groups. These baseline differences in age, race, and education may influence symptom burden and treatment responses, and they were considered in the adjusted analyses where possible.

### 3.2. Effectiveness of Personalized Exercise Programs on Symptoms, Resilience, and Quality of Life

For each outcome, all available data were included in the analyses, so the number of participants analyzed may differ between outcomes due to data completeness. Outcome-specific sample sizes varied due to missing follow-up assessments. The study evaluated the impact of exercise interventions on symptoms, resilience, and quality of life, comparing baseline (T1) and post-intervention (T2) mean scores between the exercise (n = 53) and control (n = 22) groups. The exercise group combined both in-person and telehealth participants because both delivery methods utilized identical exercise content and follow-up structure, differing only in delivery modality. The exercise group showed significant improvements in physical fatigability (*t* = 2.951, *p* < 0.006) and mental fatigability (*t* = 3.132, *p* < 0.004), with mean scores decreasing from 25.6 (SD = 15.0) to 19.8 (SD = 13.9) and from 21.8 (SD = 15.0) to 15.4 (SD = 14.2), respectively. The control group experienced a significant increase in pain levels (*t* = −3.1, *p* < 0.01), with mean scores rising from 46.6 ± 8.5 to 52.0 ± 7.7. No significant changes were observed in other symptoms, resilience, or quality of life measures (Table 3).

### 3.3. Effectiveness Comparison Between the Two Delivery Methods

An ANCOVA was conducted to compare the impact of the exercise program delivered through in-person sessions versus telehealth, adjusting for age and education (Table 4). This adjustment was made to account for significant baseline differences across groups. The mean changes (completion-baseline scores) were used for comparison. A significant difference was observed in visuo-perceptual cognitive difficulty, with a mean change of −1.1 ± 2.2 for the iHBE group and −1.5 ± 5.8 for the TEHE group (F = 3.55, *p* = 0.027), indicating that the telehealth group experienced a slightly greater reduction in visuo-perceptual difficulties. All other variables showed no statistically significant differences between in-person and telehealth groups. The findings suggest similar impacts of both delivery methods on these outcomes. It appears that both delivery methods had comparable effects across most outcomes, with the exception of a minor cognitive benefit favoring the telehealth group.

## 4. Discussion

This secondary analysis supports the effectiveness of personalized home exercise programs—whether delivered in-person or via telehealth—in improving specific outcomes for cancer survivors. Consistent with other studies [46,47,48], our findings demonstrate significant improvements in both physical and mental fatigability among participants receiving exercise interventions, particularly within the TEHE group, whereas the control group reported worsening symptoms. These improvements are likely attributable to several interacting mechanisms. Tailoring exercise recommendations to each participant’s needs and abilities may enhance self-efficacy, motivation, and adherence. Additionally, incorporating mobile technologies and bidirectional communication via smartphone applications allows for real-time feedback, individualized goal setting, and ongoing support—factors known to promote sustained physical activity and effective symptom management among cancer survivors. By combining personalized program design with technology-enabled engagement and support, these interventions may effectively address common barriers and optimize health outcomes in this population.

The comparison between in-person and telehealth delivery methods revealed no significant differences in most outcomes, indicating that telehealth is a viable alternative to in-person sessions for implementing personalized exercise interventions. These results align with previous literature, confirming the same benefits of exercise interventions delivered through telehealth; in-person are comparable for the general population [49] and other chronic illnesses such as cardiovascular disease [50], type 2 diabetes, and obesity [51]. In some populations, telehealth exercise programs even showed better health outcomes [52]. The comparable effectiveness of telehealth may be explained by several mechanisms inherent to remote delivery. Telehealth platforms can facilitate frequent, timely interactions and provide personalized feedback, educational resources, and motivational support, all of which are critical for fostering engagement and sustained behavior change [53]. The convenience and accessibility of remote interventions can also help to reduce traditional barriers to participation—such as travel, scheduling conflicts, or treatment-related fatigue—thereby enabling more consistent adherence and continued self-management [54]. Additionally, digital platforms often enable tailored monitoring and rapid responsiveness to participants’ individual progress or challenges, further enhancing the fidelity and effectiveness of interventions [55]. Given these comparable—and in some cases, superior—outcomes of telehealth exercise programs, it is recommended to consider implementing or expanding telehealth options for personalized exercise interventions, particularly for cancer survivors, as these approaches may reduce cancer-related fatigue and support recovery.

One notable finding is the greater improvement in visuo-perceptual cognitive difficulty within the telehealth group, which may indicate specific advantages of this delivery method in certain cognitive domains. While it is known that certain aerobic exercises (e.g., dance, yoga, Tai chi) can improve visuo-perceptual cognitive functions, particularly those requiring coordination and balance [56,57], it is unclear how telehealth would enhance cognitive function. It is possible that the use of smartphone applications, visual aids from videos and graphics, and weekly physical performance reports might affect visuo-perceptual skills. Furthermore, the convenience and accessibility of telehealth may reduce the stress and anxiety associated with in-person visits, thereby allowing participants to focus better on cognitive tasks. More studies are needed to verify this finding.

In interpreting these findings, it is important to consider both the successes and challenges associated with implementing personalized exercise programs, particularly through telehealth delivery. Across intervention arms, the use of individualized recommendations and ongoing technological support likely contributed to strong participant engagement and adherence, factors that are essential for achieving health benefits. Nevertheless, several barriers may have impacted implementation and intervention fidelity. Variability in participants’ digital literacy, internet access, and comfort with mobile applications could have influenced their ability to fully utilize telehealth features. Additionally, maintaining motivation and ensuring regular participation can be more challenging in remote settings, where direct personal encouragement may be less frequent or nuanced. As this is a secondary analysis, detailed data on intervention adherence, session attendance, and protocol fidelity were not systematically available, limiting our ability to rigorously assess how consistently the interventions were delivered and received.

Generalizability should be interpreted with caution in this study. Participants were recruited from pilot studies conducted at a single comprehensive cancer center, which may limit the applicability of these findings to the broader population of cancer survivors. Differences in sample characteristics—including age, race, educational background, socioeconomic status, access to technology, and cancer types—may affect the extent to which these results can be generalized. Additionally, inclusion criteria such as the requirement for smartphone ownership and the presence of moderate or greater fatigue may have excluded individuals with more severe symptoms, those with limited digital literacy, or from lower socioeconomic backgrounds. As a result, the effectiveness and feasibility of these interventions may differ in more heterogeneous, less technologically engaged, or resource-limited settings. Future studies employing larger, more diverse cohorts across multiple sites are needed to better assess the external validity and to confirm the transferability of these findings to a wider range of cancer survivor populations.

In summary, these findings reinforce and extend the growing body of literature supporting the implementation of individualized exercise interventions for cancer survivors. Both in-person and telehealth delivery methods demonstrated effectiveness in reducing cancer-related fatigability, findings that align with prior randomized controlled trials and meta-analyses showing exercise to be broadly beneficial for symptom management and quality of life in this population [54,58,59]. The comparable outcomes between telehealth and in-person approaches are consistent with recent evidence across populations and chronic diseases, indicating that remote, technology-mediated interventions can foster similar or even superior health behavior change and outcomes when grounded in established behavioral frameworks such as social cognitive theory [60,61] or the NIH Symptom Science Model [62,63]. These frameworks highlight self-efficacy, goal setting, feedback, and personalized support as critical active ingredients—mechanisms that appear present in both delivery modes of our study. However, the exploratory, secondary design, modest sample size, and possible group differences at baseline limit the strength of inferences and the ability to determine causal pathways with certainty. Nevertheless, the overall pattern of results underscores the promise and theoretical plausibility of both telehealth and in-person home-based exercise interventions as accessible, scalable strategies for post-cancer symptom management.

### 4.1. Implications

Given the exploratory and secondary nature of this analysis, the present findings should be viewed as preliminary but promising evidence for implementing both telehealth and in-person personalized exercise interventions for cancer survivors. Larger, multi-center randomized controlled trials are needed to confirm these results, clarify long-term benefits, and determine which survivor subgroups are most likely to benefit [54]. There is also a need for future research to integrate additional supportive components into exercise programs, such as mindfulness-based practices [64] and symptom management strategies like auricular acupressure [65,66,67], and to evaluate their efficacy when delivered via telehealth and conventional approaches. During the study period, there has been a rapid increase in the use of social media platforms to facilitate peer support and engagement in exercise programs [68,69]. More recently, the application of artificial intelligence to personalize health interventions has gained significant interest. Future research should explore how these evolving technologies can be combined to further enhance the effectiveness and reach of telehealth-based exercise interventions for cancer survivors.

On a practical level, healthcare programs can utilize telehealth to increase accessibility for survivors living in rural or underserved communities or those with mobility limitations [70]. The flexibility and convenience afforded by telehealth platforms may improve adherence by allowing individuals to participate from their own homes at times that best fit their schedules. Incorporating digital tools and personalized, interactive support may help address common barriers to engagement [53]. However, attention must be paid to potential disparities in digital literacy and technology access to enable equitable participation and maintain high fidelity of program delivery.

From a policy perspective, these results highlight the importance of developing and maintaining supportive telehealth infrastructure, reimbursement mechanisms, and digital literacy initiatives to ensure telehealth interventions are accessible and effective for a wide range of cancer survivors [70,71,72]. Policy strategies that embed multi-component survivorship support—including exercise and psychosocial care—within standard cancer care are likely to promote better health outcomes and greater equity across survivor populations.

### 4.2. Limitations

This comparative study has several limitations that should be considered when interpreting the findings. First, the relatively small sample sizes in each group may limit the generalizability of the results. As a pilot and exploratory secondary analysis, the study was not powered to detect small effects or adjust for a broad range of covariates. Larger, more diverse samples and longer follow-up periods are needed in future studies to confirm these findings and examine the long-term effects of personalized exercise interventions among cancer survivors.

Second, due to its secondary data analysis design, the study populations differed in several key baseline characteristics, including age, race, education level, household income, cancer diagnosis, and disease stage. For example, the iHBE group did not include participants with gastrointestinal, central nervous system, or skin cancers, and notable age distribution differences existed across groups. These demographic and clinical differences may influence symptom burden and response to interventions, thereby limiting the comparability of outcomes across different delivery modalities. We included age and education as covariates in our ANCOVA models to address potential confounding, but we excluded other variables such as marital status and income to avoid overfitting, which may have influenced our results. To enhance statistical power and analytic consistency, we combined the control groups from both studies; however, this decision may have introduced additional variability affecting interpretation.

Third, while telehealth-based delivery offers greater accessibility and flexibility, it can also introduce challenges in sustaining participant motivation and providing individualized encouragement compared to in-person sessions. These implementation issues were not assessed in the current study. Future work should include measures of participant satisfaction, engagement, and adherence to better understand the nuances and effectiveness of telehealth interventions.

Fourth, we did not collect cost data, limiting our ability to evaluate the comparative cost-effectiveness of in-person versus telehealth exercise delivery. Future studies should integrate economic evaluations to better inform policy and the scalability of survivorship interventions. We also did not measure physical activity outside the intervention protocols, such as recreational or informal exercise, which could have impacted symptom trajectories and functional outcomes. In addition, information on concurrent cancer treatments, medication use, or comorbidities was not collected, despite their potential influence on symptom experience and response to intervention. Addressing these unmeasured factors in future research will help to more accurately isolate the effects of exercise programs.

Finally, while most participants were recruited through structured channels, a small number (n = 6) enrolled via self-referral, introducing potential selection bias. Despite these limitations, the inclusion of participants with diverse backgrounds, including age, gender, race, and education, strengthens the relevance of our findings to the broader cancer survivor population. Nonetheless, more comprehensive baseline assessments and larger, more representative samples are necessary in future studies to strengthen internal validity and generalizability.

## 5. Conclusions

The results of the present study indicate that personalized exercise programs delivered through both in-person visits and telehealth effectively improved various health outcomes for cancer survivors. Specifically, significant improvements were observed in physical and mental fatigability, particularly in the telehealth group. Additionally, trends of improvement were noted in both groups’ insomnia, cognitive impairment, and visuo-perceptual tasks, although these results were not statistically significant. The comparison between in-person and telehealth delivery methods revealed no significant differences in most outcomes, suggesting that telehealth is a viable alternative to in-person sessions for personalized exercise interventions.

## Figures and Tables

**Table 1 cancers-17-02432-t001:** Summary of the sample characteristics for the two pilot studies.

Characteristics		In-Person Exercise Study (iHBE) (n = 25)	Telehealth Exercise Study (TEHE) (n = 50)
Age (mean ± SD)		73.92 ± 6.74	58.30 ± 12.43
Sex (n, %)	Male	12 (48.0)	24 (48.0)
	Female	13 (52.0)	26 (52.0)
Race (n, %)	White	9 (36.0)	41 (82.0)
	Black or African American	15 (60.0)	3 (6.0)
	Others	1 (4.0)	6 (12.0)
Education (n, %)	College/university and less	11 (44.0)	6 (12.0)
	Graduate level	11 (44.0)	41 (82.0)
	Not disclosed	3 (12.0)	3 (6.0)
Employment (n, %)	Employed	7 (28.0)	19 (38.0)
	Not employed	18 (72.0)	31 (62.0)
Marital status (n, %)	Married	8 (32.0)	38 (76.0)
	Single, divorced, or widowed	17 (68.0)	12 (24.0)
Cancer types (n, %)	Thoracic cancers	1 (4.0)	4 (8.0)
	Breast cancers	4 (16.0)	13 (26.0)
	Gastrointestinal cancers	-	8 (16.0)
	Genitourinary cancers	11 (44.0)	11 (22.0)
	Gynecologic cancers	5 (20.0)	3 (6.0)
	Central nervous system cancers	-	2 (4.0)
	Skin cancers	-	1 (2.0)
	Metastatic cancers	4 (16.0)	8 (16.0)
Baseline self-report fatigue (mean ± SD)		46.97 ± 9.82	51.23 ± 9.91

**Table 2 cancers-17-02432-t002:** Demographic and general characteristics of participants in three groups (N = 75).

Variables	Categories	In-Person Group (iHBE) (n = 15)	Telehealth Group (TEHE) (n = 38)	Control (n = 22)	χ^2^/F	*p*
Age (mean ± SD)		74.33 ± 7.62	58.61 ± 12.10	64.59 ± 13.43	9.686	<0.001
Gender (n,%)	Male	7 (46.7)	18 (47.4)	11 (50.0)	0.052	>0.009
	Female	8 (53.3)	20 (52.6)	11 (50.0)		
Race (n,%)	White	5 (33.3%)	32 (84.2%)	13 (59.1)	18.836	<0.001
	Black or African American	9 (60.0%)	2 (5.3%)	7 (31.8)		
	Others	1 (6.7%)	4 (10.5%)	2 (9.1)		
Education (n,%)	Less than college	6 (42.9%)	4 (10.8%)	11 (61.1%)	15.75	<0.001
	College graduate	8 (57.1%)	33 (89.2%)	7 (38.9%)		
Employment (n,%)	Employed	4 (26.7)	14 (37.8)	8 (36.4)	0.531	0.811
	Not employed	11 (73.3)	24 (63.2)	14 (63.6)		
Marital status (n,%)	Married	7 (46.7)	28 (73.7)	11 (50.0)	4.996	0.092
	Single, divorced, or widowed	10 (26.3)	11 (50.0)	8 (53.3)		

**Table 3 cancers-17-02432-t003:** Comparison of baseline and post-intervention mean score of outcomes between exercise and control groups (N = 75).

Variables			Exercise Groups (In-Person + Telehealth) (n = 53)			Control Group (n = 22)		
			Mean ± SD		*t*	Mean ± SD		*t*
			Baseline	Completion		Baseline	Completion	
**Pain**			48.1 ± 9.2	49.7 ± 10.8	−0.99	46.6 ± 8.5	52.0 ± 7.7	−3.1 **
**Fatigue**			50.0 ± 11.0	50.1 ± 10.5	−0.08	46.5 ± 8.7	50.2 ± 9.4	−1.5
**Fatigability**	Physical		25.6 ± 15.0	19.8 ± 13.9	3.0 **	18.8 ± 19.1	19.3 ± 17.0	−0.13
	Mental		21.8 ± 15.0	15.4 ± 14.2	3.1 **	11.5 ± 13.1	12.8 ± 12.0	−0.39
**Sleep**	Insomnia severity score		16.8 ± 6.2	14.9 ± 5.0	1.9	17.8 ± 4.8	17.1 ± 6.8	0.57
	Sleep time (hr.)		7.2 ± 1.1	7.3 ± 1.4	−1.2	6.8 ± 1.1	7.4 ± 1.4	−1.7
**Cognitive function**	MoCA ^1^ score		27.2 ± 1.9	26.9 ± 2.0	0.8	24.7 ± 4.3	25.9 ± 3.6	−1.7
	MASQ ^2^	Language	11.4 ± 4.5	11.3 ± 5.0	0.34	13.0 ± 6.0	13.6 ± 6.2	−0.82
		Visuo-perceptual	9.2 ± 5.0	7.8 ± 3.7	1.6	10.5 ± 5.6	11.1 ± 5.1	−0.84
		Verbal memory	16.8 ± 4.4	17.1 ± 4.4	−0.37	16.4 ± 4.9	16.8 ± 6.1	−0.35
		Visual memory	15.7 ± 4.2	15.6 ± 4.3	0.12	14.4 ± 4.4	15.5 ± 3.9	−1.7
		Attention	18.0 ± 4.2	16.9 ± 4.4	1.9	15.4 ± 5.0	15.3 ± 4.5	0.15
**Physical function**	vSPPB score ^3^		9.3 ± 2.3	9.7 ± 2.5	−1.0	8.3 ± 3.1	7.6 ± 2.4	1.2
	Average daily steps over 7 days (×1000),		6.3 ± 3.3	6.7 ± 4.3	−0.71	5.5 ± 2.3	4.5 ± 2.3	1.8
**Resilience**			31.8 ± 6.4	32.1 ± 5.6	−0.44	36.7 ± 7.4	37.4 ± 7.8	−0.49
**Quality of Life**	Physical component		50.8 ± 10.2	49.6 ± 10.8	0.79	52.6 ± 9.2	49.7 ± 8.0	1.56
	Mental component		49.7 ± 10.3	49.5 ± 10.1	0.15	53.1 ± 6.8	50.0 ± 10.3	1.40

^1^ MoCA, Montreal Cognitive Assessment; ^2^ MASQ, Multiple Ability Self-Report Questionnaire; ^3^ vSPPB, Virtual Short Physical Performance Battery; ** *p* ≤ 0.01.

**Table 4 cancers-17-02432-t004:** Differences of mean change (T2-T1) of all outcomes between the in-person and telehealth groups (N = 53), controlled for age and education.

Variables			Mean Change ± SD		F	*p*
			In-Person (n = 15)	Telehealth (n = 38)		
**Pain**			2.0 ± 11.1	1.4 ± 8.6	0.21	0.888
**Fatigue**			1.0 ± 7.6	−0.2 ± 9.8	0.25	0.862
**Fatigability**	Physical		0.3 ± 8.3	−7.6 ± 11.1	1.03	0.394
	Mental		−0.6 ± 10.5	−8.1 ± 11.2	1.17	0.339
**Sleep**	Insomnia severity score		−1.4 ± 8.4	−2.0 ± 5.0	0.64	0.595
	Sleep time (minutes)		12.9 ± 37.9	7.2 ± 43.6	0.644	0.430
**Cognitive function**	MoCA ^1^ score		−1.4 ± 2.6	0.1 ± 2.2	1.40	0.265
	MASQ ^2^	Language	0.3 ± 3.8	−0.3 ± 2.0	0.19	0.904
		Visuo-perceptual	−1.1 ± 2.2	−1.5 ± 5.8	3.55	0.027
		Verbal memory	1.1 ± 7.6	0.4 ± 4.0	0.39	0.760
		Visual memory	0.1 ± 1.6	−0.1 ± 3.2	0.66	0.584
		Attention	−1.2 ± 4.1	−1.1 ± 3.2	0.02	1.000
**Physical function**	vSPPB score ^3^		−0.4 ± 1.8	0.6 ± 1.7	0.60	0.622
	Average daily steps		185.2 ± 3148.1	418.4 ± 2544.4	0.28	0.842
**Resilience**			−1.1 ± 4.5	0.7 ± 4.4	2.02	0.133
**Quality of Life**	Physical component		0.7 ± 9.5	−1.9 ± 6.8	0.41	0.745
	Mental component		−1.0 ± 4.9	0.56 ± 8.5	0.61	0.613

^1^ MoCA, Montreal Cognitive Assessment; ^2^ MASQ, Multiple Ability Self-Report Questionnaire; ^3^ vSPPB, Virtual Short Physical Performance Battery.

## Data Availability

The data that support the findings of this study are available from the corresponding author upon reasonable request.

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
