# Peer review of "A Comparison of In-Person and Telehealth Personalized Exercise Programs for Cancer Survivors: A Secondary Data Analysis"

_cancers, 2025, doi:10.3390/cancers17152432_

Round 1

Reviewer 1 Report

Comments and Suggestions for Authors

This study assessed the impact of a personalized exercise program on cancer-related symptoms, resilience, and quality of life, comparing in-person and telehealth delivery methods. Both approaches led to improvements in physical and mental fatigability. No major differences were found between delivery methods, except for slight cognitive benefits in the telehealth group. Findings suggest telehealth may be a viable alternative to in-person interventions, warranting further research.

Although the research presented in this manuscript has relevance, there are some issues that the authors need address to make their paper publishable. Major shortcomings are listed as follows.

The introduction section is generally clear and well-structured. However, there are a few areas that should be improved for clarity, precision, and academic rigor.

The authors state that “little research has been conducted” on comparing delivery methods. However, their statement remains vague. More specificity and further referencing about the lack of comparative studies would strengthen this point.

The authors should provide a more in-depth theoretical framework for their study. For instance, the introduction could briefly mention why or how exercise improves symptoms or HRQOL - mentioning possible physiological or psychological mechanisms would enhance depth. Additionally, concepts like “resilience” and “HRQOL” are mentioned in the objectives but not explained or contextualized in the introduction section. A brief description would help frame their relevance.

Although the authors clearly outline the objectives of the study - evaluating the effects of personalized exercise interventions and comparing telehealth versus in-person delivery - they do not explicitly state testable hypotheses. Formulating hypotheses (e.g., “Participants receiving exercise interventions will show greater improvements in fatigue and HRQOL compared to controls” or “Telehealth delivery will be equally effective as in-person delivery”) would have provided a clearer framework for the study’s analytical approach and allowed readers to better understand the expected direction of effects. Including hypotheses also strengthens the scientific rigor of a study by clarifying its predictive stance and aligning the results with expectations.

The two comparison samples are not perfectly comparable in terms of cancer types. For example, the 'In-person exercise' (iHBE ) sample does not include patients with gastrointestinal cancers, central nervous system cancers, or skin cancers. Moreover, the proportion of patients with common cancer types is not always proportional to the size of each specific sample. There are also differences in terms of age. That may affect results. The authors should provide arguments about this issue. Please, take into account also the adoption of the control groups from both studies (line 210) as a strategy to analyse results. See also the limitations emphasized in Section 4.2.

I suggest the authors to compare both approaches - “in-person” and “telehealth” - in terms of management costs. That would increase the relevance of the study from a healthcare policy perspective.

Reviewer 2 Report

Comments and Suggestions for Authors

The authors present an interesting study in which two interventions designed to assist patients who have completed cancer-related therapies and entered remission carry out exercise regimes are examined with respect to one another, and also patients not in receipt of either. Briefly, each intervention is wholly distinct from the other, with one involving home visits from someone who is there to guide and motivate and support the individual over the duration of the program, while the other utilises a smart phone app that designs and prompts the individual to meet exercise goals set by the application itself. Of those who partook in the study, those who received either the ‘home’ intervention or the ‘smart phone app’ intervention demonstrated improved scores in categories such as physical health, mental fatigability and others as compared to those individuals not undertaking any regimes. In comparing both to one another, there was very few differences observed, highlighting the strengths of both with respect to one another. Overall, this was an interesting read that is well put together, and balanced in its strengths and limitations accordingly.

In reviewing the manuscript I made a couple of observations. The following should be considered by the authors when preparing a suitable revision.

  1. Was it taken into account whether any of the participants took part in hobbies or sporting activities outside of the programs being evaluated?
  2. I think it should be made clearer as to whether the participants were still in receipt of treatment/medications that might impact on some of the scales used. This might be related to the cancer diagnosis, or perhaps even a comorbidity.
  3. For the ‘in-person home exercise program’ was suitable training given to the individuals who were conducting the home visits? Was there any efforts made to ensure perhaps the same person visited the same participants over the duration of the program, or was the visit random depending on person availability?
  4. The formatting of the data in the tables requires attention. The font style changes frequently – the authors should revise this and ensure it is one single font style throughout.  

Reviewer 3 Report

Comments and Suggestions for Authors

The introduction offers a rationale for the investigation (especially related to the Covid era)  and ends with a research question.

In the section methods: the self-referral might introduce a bias (line 118-9). How many were these? 

The use of the measurement scalesand the statistical analysis seems adequate

The patient numbers are low (in spite of a 4 year recruitment period) which limites the applicability of the results. This has been acknowledged as a limitation. 

Table 1 shows large differences in sociodemographic variables which might affect outcome. Has the marital status ora education, to name some examples been taken into account toe explain the resuslts? This information is vital. 

Comments on the Quality of English Language

no specific comments 

Reviewer 4 Report

Comments and Suggestions for Authors

The COVID-19 pandemic changes our view of life and social communication style. Moreover, the pandemic has had a significant impact on the healthcare system. Fortunately, the above situation brought us the new possibility of artificial intelligence applications and social media's usefulness for health protection and diagnosis. Therefore, the article entitled A Comparison of In-Person and Telehealth Personalized Exercise Program for Cancer Survivors: A Secondary Data Analysis discloses the differences and pros and cons of the two styles of communication. Authors, by their studies, have shown the effectiveness of a telehealth personalised exercise on fatigability and cognitive difficulty. Even though the investigated group was not very numerous, the results are worth publication.

The study has been well presented. The manuscript was well written and readable, with correctly selected and cited references.

Round 2

Reviewer 1 Report

Comments and Suggestions for Authors

The authors have satisfactorily addressed my concerns, and I believe the paper has been significantly improved in clarity, methodological rigor, and theoretical contribution.

Reviewer 3 Report

Comments and Suggestions for Authors

The low numbers of included patients and consequenyly the inability to take into account social and marital status in the analysis remains a serious problem.